# Learning to Infer the Structure of Network Games

## Abstract

Strategic interactions between a group of individuals or organisations can be modelled as games played on networks, where a player's payoff depends not only on their actions but also on those of their neighbors. Inferring the network structure from observed game outcomes (equilibrium actions) is an important problem with numerous potential applications in economics and social sciences. Existing methods mostly require the knowledge of the utility function associated with the game, which is often unrealistic to obtain in real-world scenarios. To address this limitation, we propose a novel transformer-like architecture which correctly accounts for the symmetries of the problem and learns a mapping from the equilibrium actions to the network structure of the game without explicit knowledge of the utility function. We test our method on three different types of network games using both synthetic and real-world data, and demonstrate its effectiveness in network structure inference and superior performance over existing methods.

## 1 Introduction

Individuals or organizations cooperate with or compete against each other in a wide range of practical situations. Strategic interactions between individuals are often modeled as games played on networks, where an individual's utility depends not only on their actions (and characteristics) but also on those of their neighbors. In a network game (and games in general), the utility function is of fundamental importance, as rational agents maximize their utilities when making real-life decisions. The current literature on network games has primarily focused on the scenarios where the utility function is predefined, and the structure of the network (represented by a graph) is known beforehand. However, in practical settings, while it is common to observe the actions of the players, the underlying interaction network and the utility function remain hidden. This makes it challenging to exploit network information and utility function for behavioral predictions and network-based interventions (e.g., marketing campaigns or information diffusion).

In this paper, we focus on the problem of inferring the structure of the interaction network from observed equilibrium actions of a network game. A few recent studies have tackled similar problems (Irfan & Ortiz, 2011; Honorio & Ortiz, 2015; Ghoshal & Honorio, 2017a;b; Garg & Jaakkola, 2016; 2017; Barik & Honorio, 2019; 2020; Leng et al., 2020); however, several major limitations remain. First, all these methods require the explicit knowledge of the utility function to infer the underlying network structure, which may be impractical to assume and may also change over time. The work of Honorio & Ortiz (2015) considers a more general hypothesis space of games for linear influence game, but they only focus on binary actions and linear payoffs. Second, the methodology in each of these studies has been designed for the specific game under consideration, thereby limiting its scope in handling a wide range of strategic interactions in real-world scenarios.

To address these limitations, we first summarise three common network games studied in the recent literature in a generic form, which is based on both individual and network factors that impact one's utility and the corresponding equilibrium actions. Despite the different nature of the games, it permits the relation between the equilibrium actions and network structure to be written in a unified manner. This motivates us to propose a data-driven model where we learn the functional mapping between the equilibrium actions and network structure *without* explicit knowledge about the utility function of the game. Our model is based on an encoder-decoder approach where the encoder is a transformer architecture and the decoder can be chosen flexibly, and is trained using pairs of actions and network

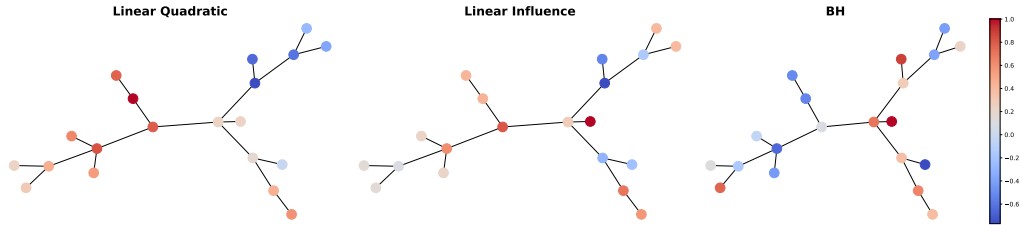

Figure 1: Example actions for different types of games on a Barabási-Albert graph. Actions are normalized to be in $[-1, 1]$ and are displayed as colors on nodes.

structures. Once trained, the model can be deployed to infer the network structure from only observed actions. Synthetic and real-world experiments demonstrate the superiority of our method against several state-of-the-art approaches.

**Main contributions.** First, we compare three network games using a unified parameterization, which helps reveal the different nature of these games and interpret the strategic interactions they represent. Second, to our knowledge, our framework is one of the first to infer the network structure behind the games without knowing the utility function. This capability is important in real-world scenarios where the nature of the interactions remains hidden or may even evolve over time. Finally, our work contributes to the emerging field of data-driven structural inference by proposing a model based on a novel permutation-invariant transformer architecture. Such a framework may find numerous applications in diverse domains as transportation, social, biological, and financial network analysis.

## 2 SETTING

In this section, we start by analysing three commonly studied network games (Sections 2.1 and 2.2). Based on the specific utility function of these games, we establish a generic relationship between the equilibrium actions and network structure (Section 2.3), which eventually motivates the proposed framework that learns network structure without any knowledge of the utility function (Section 3).

### 2.1 CONTINUOUS-ACTION NETWORK GAMES

Consider a network of $N$ individuals represented by a weighted and undirected graph $\mathcal{G}(\mathcal{V}, \mathcal{E})$, where $\mathcal{V}$ and $\mathcal{E}$ denote the node and edge sets, respectively. For any pair of individuals $i$ and $j$, $w_{ij} = w_{ji} > 0$ if $(i, j) \in \mathcal{E}$ and $w_{ij} = w_{ji} = 0$ otherwise, where $w_{ij}$ is the $ij$-th entry of the adjacency matrix $\mathbf{W}$. In this work, we assume the graph $\mathcal{G}$ is connected, and consider the normalized adjacency matrix $\mathbf{A} = \mathbf{D}^{-\frac{1}{2}}\mathbf{W}\mathbf{D}^{-\frac{1}{2}}$ where $\mathbf{D} = \text{diag}(\mathbf{W1})$, where $\mathbf{1}$ is an $N$-dimensional vector of ones. In a network game, the payoff $u_i$ of an individual $i$ depends on their action $x_i$ as well as the actions of neighbors $j \in \mathcal{N}_i$. We consider three commonly studied network games: linear quadratic games (Ballester et al., 2006), a variation of the linear influence games developed in Irfan & Ortiz (2011), and the graphical game studied in Barik & Honorio (2019) (which we will refer to as Barik-Honorio or BH graphical game).

**Linear quadratic games.** Linear quadratic games are widely studied in the economics literature (Jackson & Zenou, 2014; Ballester et al., 2006). In this game, a player $i$ chooses their action by maximizing the following utility function:

$$\max_{\{x_i\}} u_i = b_i x_i - \frac{1}{2}x_i^2 + \beta \sum_{j \in \mathcal{N}_i} a_{ij} x_i x_j, \tag{1}$$

where $b_i$ represents the marginal benefit of $i$ by taking action $x_i$ and $\beta$ is the strength of dependencies between actions of neighbors in the network, respectively. Note that this utility function can also be thought of as a second-order approximation to non-linear utility functions of more complex games. The pure-strategy Nash equilibrium (PSNE) of this game is

$$\mathbf{x}^* = (\mathbf{I} - \beta\mathbf{A})^{-1}\mathbf{b}, \tag{2}$$

where $\mathbf{x}^*$ and $\mathbf{b}$ are $N$-dimensional vectors collecting actions and marginal benefits for all individuals, and $\mathbf{I}$ is the $N \times N$ identity matrix. Under the assumption that $|\beta| < 1$, the matrix inverse is guaranteed to exist as the spectral radius of $\mathbf{A}$ is 1. Furthermore, when $\beta > 0$, the game corresponds to a strategic complement relationship (i.e., intuitively, the incentive of a player to take a higher action is increasing in the number of their neighbors also taking a higher action); when $\beta < 0$, it corresponds to strategic substitute (i.e., intuitively, the incentive of a player to take a higher action is decreasing in the number of their neighbors also taking a higher action).

**Linear influence games.** Inspired by the threshold model (Granovetter, 1978), Irfan & Ortiz (2011) proposed the linear influence games, where an individual chooses the action that maximizes the following utility function[1]:

$$\max_{\{x_i\}} u_i = \sum_{j \in \mathcal{N}_i} a_{ij} x_i x_j - b_i x_i, \tag{3}$$

where $b_i$ can be understood as a threshold parameter for $i$'s level of tolerance for negative effects. Under the assumption that $\mathbf{A}$ is invertible, the PSNE satisfies the following condition:

$$\mathbf{x}^* = \mathbf{A}^{-1} \mathbf{b}. \tag{4}$$

**BH graphical game.** Finally, in the specific graphical game introduced in Barik & Honorio (2019), an individual maximizes the following utility function:

$$\max_{\{x_i\}} u_i = - \left\| x_i - \sum_{j \in \mathcal{V}} a_{ij} x_j \right\|_2. \tag{5}$$

This utility can be used to model games where an individual prefers to conform to the social norm (actions of their neighbors), since their own utility decreases as their action deviates from those of their neighbors. The PSNE for this game satisfies the following condition:

$$\mathbf{x}^* = \mathbf{A} \mathbf{x}^*. \tag{6}$$

This suggests that $\mathbf{x}^*$ is the eigenvector $\mathbf{u}_1$ of $\mathbf{A}$ which is associated with the largest eigenvalue (which is 1). We consider actions from the set of $\epsilon$-PSNE (Barik & Honorio, 2019), which are obtained by adding noise independently per player. The observed actions $\mathbf{x}$ are: $\mathbf{x} = \mathbf{x}^* + \mathbf{e}$, where $\mathbf{e}$ is sub-Gaussian noise[2].

Table 1: Parameterization of three network games considered in the paper.

|  | $\mathcal{F}(\mathbf{A})$ | $\mathcal{H}(\mathbf{b})$ |
|---|---|---|
| Linear quadratic | $(\mathbf{I} - \beta \mathbf{A})^{-1}$ | $\mathbf{b}$ |
| Linear influence | $\mathbf{A}^{-1}$ | $\mathbf{b}$ |
| Barik-Honorio | $\mathbf{u}_1$ | 1 |

By investigating equations equation 2, equation 4, and equation 6, we can write the condition for the equilibrium actions $\mathbf{x}^*$ in a generic form (we tacitly assume that $|\beta| < 1$ and $\mathbf{A}$ is invertible, see Table 1):

$$\mathbf{x}^* = \mathcal{F}(\mathbf{A})\mathcal{H}(\mathbf{b}), \tag{7}$$

where $\mathcal{F}(\mathbf{A})$ is a function of the network structure and $\mathcal{H}(\mathbf{b})$ is a function of additional parameters (if any) associated with the game. That is, $\mathcal{F}(\mathbf{A})$ accounts for the influence from the actions of one's neighbors in the network. Conversely, $\mathcal{H}(\mathbf{b})$ is only affected by one's idiosyncratic (individual) characteristics.

## 2.2 MODELING OF INDIVIDUAL IDIOSYNCRATIC CHARACTERISTICS

Under the linear quadratic or linear influence games, the parameter $\mathbf{b}$ captures the marginal benefits or tolerance levels as idiosyncratic characteristics of players in the corresponding games. In the presence of the homophily effect (McPherson et al., 2001), we may assume that this parameter is associated with the network structure. To this end, we propose to model $\mathbf{b}$ as follows:

$$\mathbf{b} \sim \mathcal{N}(\mathbf{0}, \mathbf{L}_\alpha^\dagger), \qquad \mathbf{L}_\alpha = (1 - \alpha)\mathbf{I} + \alpha \mathbf{L} \tag{8}$$

where $\mathbf{L} = \mathbf{I} - \mathbf{A}$ is the normalized graph Laplacian matrix, and $\dagger$ represents pseudoinverse in case the matrix is not invertible (this happens when $\alpha = 1$ as $\mathbf{L}$ has a smallest eigenvalue of 0). The parameter $\alpha \in [0, 1]$ controls the relation of the individual idiosyncratic characteristics to the network structure. The two corner cases $\alpha = 0$ (for which we have $\mathbf{L}_0 = \mathbf{I}$), and $\alpha = 1$ (when $\mathbf{L}_1 = \mathbf{L}$) correspond to independent idiosyncratic characteristics and homophilous idiosyncratic characteristics (individuals with similar characteristics tend to be connected), respectively. By varying $\alpha$ from 0 to 1 we can achieve increasing levels of homophily.

---

[1] The actions are discrete in the originally proposed game. We adapt the game to a continuous setting.

[2] Sub-Gaussian variables include Gaussian variables, any bounded random variable (e.g. Bernoulli, uniform), any random variable with strictly log-concave density, and any finite mixture of sub-Gaussian variables.

## 2.3 Analysis of Equilibrium Actions

With the conditions for equilibrium actions in Section 2.1 and the modeling of individual idiosyncratic characteristics in Section 2.2, we can analyze explicitly the characteristics of these actions.

**Linear quadratic games.** Assuming $\mathbf{b}$ of the form equation 8 and using Eq. equation 2 and $\mathbf{L} = \mathbf{I} - \mathbf{A}$, we have that the equilibrium actions $\mathbf{x}^*$ follow a multivariate Gaussian distribution:

$$\mathbf{x} \sim \mathcal{N}\Big(\mathbf{0}, (\mathbf{I} - \beta\mathbf{A})^{-1}(\mathbf{I} - \alpha\mathbf{A})^{\dagger}(\mathbf{I} - \beta\mathbf{A})^{-1}\Big) = \mathcal{N}\Big(\mathbf{0}, \mathbf{U}[(\mathbf{I} - \beta\mathbf{\Lambda})^2(\mathbf{I} - \alpha\mathbf{\Lambda})]^{\dagger}\mathbf{U}^{\top}\Big), \quad (9)$$

with the eigendecomposition $\mathbf{A} = \mathbf{U}\mathbf{\Lambda}\mathbf{U}^{\top}$. Eq. equation 9 illustrates the relationship between the actions $\mathbf{x}$ and the network structure $\mathbf{A}$ and motivates the learning framework proposed in the next section. Furthermore, the covariance in Eq. equation 9 may be interpreted as a graph filter whose frequency response $\frac{1}{(1-\beta\lambda)^2(1-\alpha\lambda)}$ may shed light on the behavior of the actions. For $\beta \to 1$ and $\alpha \to 1$, the action vector $\mathbf{x}$ tends to behave like the leading eigenvectors of $\mathbf{A}$ ('low frequency'), which are smooth on the graph[3]. An example of the filter response (with $\beta = 0.8$ and $\alpha = 0.8$) applied on the eigenvalues of an instance of a 20-node Erdős–Rényi graph is shown in Fig. 2 (blue). This shows that in linear-quadratic games the actions are dominated by the leading eigenvector of A which is smooth.

**Linear influence games.** Similarly, from Eqs. equation 4 and equation 8, we have that the equilibrium actions $\mathbf{x}$ follow a multivariate Gaussian distribution:

$$\mathbf{x}^* \sim \mathcal{N}\Big(\mathbf{0}, \mathbf{A}^{-1}(\mathbf{I} - \alpha\mathbf{A})^{\dagger}\mathbf{A}^{-1}\Big) = \mathcal{N}\Big(\mathbf{0}, \mathbf{U}[\mathbf{\Lambda}^2(\mathbf{I} - \alpha\mathbf{\Lambda})]^{\dagger}\mathbf{U}^{\top}\Big). \quad (10)$$

Interpreting Eq. equation 10 as a spectral filter of the form $\frac{1}{\lambda^2(1-\alpha\lambda)}$ we can also conclude that an $\alpha \to 1$ leads to smoother actions on the graph. However, the behavior of actions in this case depends on the magnitude of the eigenvalue of $\mathbf{A}$ closest to 0. Given that the spectrum of $\mathbf{A}$ lies in the range of $[-1, 1]$, the actions are likely to behave like mid-spectrum eigenvectors, which are not necessarily smooth signals on the graph. Similarly, this can be seen from the filter response, under $\alpha = 0.8$, shown in Fig. 2 (red). This shows that in linear influence games the actions are dominated by mid-spectrum eigenvectors which are not necessarily smooth

**BH graphical game.** We see from Eq. equation 6 that the equilibrium actions correspond to the largest eigenvector $\mathbf{u}_1$ of $\mathbf{A}$. Although in this setting the observed actions are $\epsilon$-PSNE, they will still be quite smooth on the graph.

In summary, our analysis in this section shows that the smoothness of the equilibrium actions on the graph depends on $\beta$ and $\alpha$ in the linear quadratic game. The actions in linear influence game are likely to be nonsmooth, whereas those in the BH graphical game are likely to be smooth. We empirically validate this analysis in Section 5.1. Regardless of the smoothness, the relationship between the equilibrium actions and the network structure demonstrated in this section motivates us to propose a learning framework in the next section to infer network the structure from the observed actions.

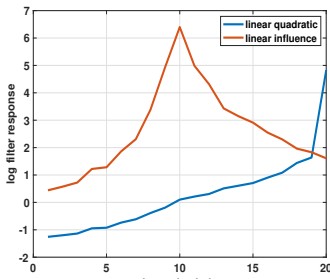

Figure 2: Interpreting actions of linear quadratic/influence games: the equilibrium actions for the two games are dominated by different sets of eigenvectors.

## 3 Proposed Approach

Motivated by the analysis in Section 2.3, we propose a model that learns a direct mapping from the observed actions to the network structure. Such a model is agnostic to the utility function of the game and avoids strong assumptions on the utility function.

In our setting, we assume to have a training set $\mathcal{D}$ of action-graph pairs $(\mathbf{X}^{(\ell)}, \mathbf{A}^{(\ell)})$ coming from games with the same (but unknown) utility function. The model takes as input an $N \times K$ matrix $\mathbf{X}$ containing the actions of $N$ players over $K$ independent games as columns, and outputs a predicted $N \times N$ adjacency matrix $\hat{\mathbf{A}} = g_{\Theta}(\mathbf{X})$ for the network game. Once trained, our model can then

---

[3]By 'smoothness' here we understand the Dirichlet energy of the features, $\mathrm{trace}(\mathbf{X}^{\top}\mathbf{L}\mathbf{X})$.

infer the network structure corresponding to previously unseen actions, as long as they are generated following a similar utility function. Moreover, this framework makes our model generalizable to learning networks with the number of nodes different from what was observed during training.

The model parameters are learned such that they minimize the cross entropy between the binary ground truth adjacency matrix and the predicted continuous one:

$$L(\Theta) = \frac{1}{|\mathcal{D}|} \sum_{(\mathbf{A}^{(\ell)}, \mathbf{X}^{(\ell)}) \in \mathcal{D}} \sum_{i,j=1}^{N} a_{ij}^{(\ell)} (\log(g_\Theta(\mathbf{X}^{(\ell)})))_{ij} + (1 - a_{ij}^{(\ell)})(1 - \log(g_\Theta(\mathbf{X}^{(\ell)})))_{ij}. \quad (11)$$

Binary cross-entropy is a standard loss function for link prediction tasks using graph neural networks (Hamilton et al., 2017), of which our problem is an instance. We tried weighting the loss based on the proportion of edges in the graph, but it did not change the performance of the model.

Our model follows an encoder-decoder architecture $g_\Theta(\mathbf{X}) = \text{dec}(\text{enc}(\mathbf{X}))$, which is a standard solution for link-prediction problems in graph neural networks (Kipf & Welling, 2016) since the resulting number of parameters of the model is independent on the size of the graph, allowing for more statistical efficiency, and for the same model to work on graphs of different sizes. The *encoder* outputs an $N \times K \times F$ tensor $\mathbf{Z} = \text{enc}(\mathbf{X})$, mapping each node $i$ in game $k$ to an $F$-dimensional latent embedding $\mathbf{z}_{ik}$. The *decoder* outputs the predicted $N \times N$ adjacency matrix $\hat{\mathbf{A}} = \text{dec}(\mathbf{Z})$, where the score $\hat{a}_{ij}$ for each edge $(i, j)$ is computed using the $K \times F$-dimensional embeddings $\mathbf{Z}_i, \mathbf{Z}_j$ of the respective nodes.

Given that the ordering of nodes in the graph is arbitrary, the model has to be a *permutation-equivariant* (Bronstein et al., 2021) function over the set of nodes: $g_\Theta(\mathbf{P}_1\mathbf{X}) = \mathbf{P}_1 g_\Theta(\mathbf{X})\mathbf{P}_1^\top$, where $\mathbf{P}_1$ is an $N \times N$ permutation matrix interpreted as reordering of the nodes of the graph. Additionally, in the case where there is no correspondence between game $k$ in one graph and game $k$ in another one (e.g. actions corresponding to user rating where however users in different graphs have rated different items), the model should also be *permutation-invariant* over the set of games, i.e. $g_\Theta(\mathbf{X}\mathbf{P}_2) = g_\Theta(\mathbf{X})$, where $\mathbf{P}_2$ is another $K \times K$ permutation matrix on the games. Overall, this combined symmetry condition can be written as $g_\Theta(\mathbf{P}_1\mathbf{X}\mathbf{P}_2) = \mathbf{P}_1 g_\Theta(\mathbf{X})\mathbf{P}_1^\top$.

**Encoder** The input to our model is a variable-length set (of actions), which excludes using a multi-layer perceptron (since it would not be able to handle the variable length) and sequence models such as LSTMs (Hochreiter & Schmidhuber, 1997) or GRUs (Cho et al., 2014) (since they treat the input as an ordered sequence, while it is an un-ordered set). Moreover, since the ground truth graph between the players is not known a priori, but it is however important for the model to exchange information between players (as the values of agents' actions are meaningless if not compared to the ones of the others), the encoder needs to perform message passing on the fully connected graph of players. Therefore, we propose a Transformer-like (Vaswani et al., 2017) encoder (which we refer to as Network Game Transformer or *NuGgeT*). *NuGgeT* processes a *set* of games happening on a given network and outputs for each node a *set* of $K$ different game-specific embeddings (one for each game that happens on the network). In this architecture, the value of action $x_{ik}$ of player $i$ in game $k$ is never mixed with the actions of other players in a different game $k'$, if not for the computation of an aggregated attention score $\alpha_{ij}$ that captures the overall similarity between players. Intuitively, in settings where we are given the outcomes of multiple games played on multiple graphs, there is no correspondence between such games. Thus, the value of an action does not bring any information about the role of a node in a graph and the only useful information that needs to be exchanged across games is the similarity of the nodes.

Specifically, for each node $i = 1, \dots, N$ and game $k = 1, \dots, K$, the embedding $\mathbf{z}_{ik}$ is computed as follows:

$$\mathbf{y}_{ik} = \text{ReLU}(x_{ik}\boldsymbol{w} + \mathbf{b}) \qquad\qquad \boldsymbol{w}, \mathbf{b} \in \mathbb{R}^F$$

$$\alpha_{ij}^{(h)} = \text{softmax}_j \Big( \bigsqcup_{k=1}^{K} \mathbf{y}_{ik}^\top \mathbf{W}_Q^{(h)} \mathbf{W}_K^{(h)} \mathbf{y}_{jk} \Big) \qquad \mathbf{W}_Q^{(h)} \in \mathbb{R}^{F \times F'}, \mathbf{W}_K^{(h)} \in \mathbb{R}^{F' \times F}$$

$$\mathbf{z}_{ik} = \psi \Big( \mathbf{y}_{ik}, \sum_{j=1}^{N} \alpha_{ij}^{(1)} \mathbf{y}_{jk}, \dots, \sum_{j=1}^{N} \alpha_{ij}^{(H)} \mathbf{y}_{jk} \Big) \qquad\qquad\qquad (12)$$

where $\square$ denotes a general permutation-invariant aggregation operator (e.g $\max$, mean $\frac{1}{K}\sum_{k=1}^{K}$ or sum $\sum_{k=1}^{K}$), $\psi$ is some learnable function, $\boldsymbol{w}, \mathbf{b}, \mathbf{W}_K^{(h)}$, and $\mathbf{W}_Q^{(h)}$ are learnable parameters, and

$h = 1, \ldots, H$ denotes the attention heads. For a generic node $i$ in game $k$, NuGgeT first expands $i$'s action $x_{ik}$ into a vector $\mathbf{y}_{ik}$ of $F$ features, it then computes $H$ attention scores (one for each head) for each pair of nodes $(i, j)$ via multiplicative attention on the expanded actions $\mathbf{y}_{\cdot k}$[4] (aggregating the unnormalized scores across games) and finally refines vector $\mathbf{y}_{ik}$ via a function $\psi$ that processes the $H$ aggregated representation of the neighbors obtained from the attention mechanism.

**Decoder**   In the case of directed graphs, there is no specific requirement for the decoder. If the graph is undirected, and therefore the adjacency matrix is symmetric, the decoder should be symmetric w.r.t. pairs of nodes (i.e., $(i, j)$ and $(j, i)$ treated the same way). Additionally, we impose invariance w.r.t. ordering of the games. *NuGgeT*'s decoder computes the predicted adjacency $\hat{a}_{ij}$ by aggregating the game-specific embeddings computed in (12) for a pair of nodes $(i, j)$ using a general permutation-invariant aggregation operator $\boxed{\cdot}$, which is then passed through a learnable function $\psi$:

$$\hat{a}_{ij} = \psi\Big( \boxed{\cdot}_{k=1}^{K} \mathbf{z}_{ik} \odot \mathbf{z}_{jk} \Big).$$

(13)

Here $\odot$ denotes element-wise product, whose use ensures symmetry w.r.t. node pairs (since $\mathbf{z}_{ik} \odot \mathbf{z}_{jk} = \mathbf{z}_{jk} \odot \mathbf{z}_{ik}$), while the permutation invariant operator over $k$ ensures invariance to the ordering of games. We empirically observe this approach to work better compared to simpler permutation-invariant functions such as the dot product of the concatenation of the games embeddings. In the SM we prove that *NuGgeT* satisfies the symmetry conditions outlined above.

## 4   RELATED WORK

*Network games*, a class of problems in game theory, have been studied extensively in computer science and economics. The majority of works in the network game literature study the characteristics of games on a known and static graph (Ballester et al., 2006; Bramoullé et al., 2014). While these studies are useful in understanding collective actions and designing interventions (Galeotti et al., 2017), it is increasingly acknowledged that networks are difficult to obtain in practice. Furthermore, the utility function associated with the game is usually unknown as well. We are interested in the *inverse problem* of inferring the network structure based on observed actions. This inverse setting is related to *graph* or *network inference*, a problem that has attracted interest in statistics (Koller & Friedman, 2009; Friedman et al., 2008a), physics (Gomez-Rodriguez et al., 2010; Gomez-Rodriguez et al., 2011), signal processing (Mateos et al., 2019; Dong et al., 2019). Our study differs from these works in accounting for the strategic interactions and the game theoretical framework underlying the observed data.

Deep learning models have been recently proposed for latent graph inference, e.g. structural learning problem to infer a DAG from the observed actions (Yu et al., 2019; Zheng et al., 2020). However, these approaches are limited to predicting acyclic graphs, whereas the graphs we are interested in are often cyclic. Kipf et al. (2018) proposed a graph neural network (GNN) model to infer the interactions of an underlying dynamical system from observational data. Differently from this work, their model is trained on predicting the future state of the system, rather than the network interactions themselves. In latent graph learning (Cosmo et al., 2020a;b; Wang et al., 2019), the graph is learnt jointly with a downstream task, conversely to our scenario where there is no downstream task. The method of Belilovsky et al. (2017) is the mostly related to ours, since they propose a supervised model to infer an undirected graphical model from observed covariates using a series of dilated convolutions. However, their model is not permutation-equivariant w.r.t. the order of nodes and the number of layers depends on the number of nodes. Both issues cause the statistical efficiency of the model to scale poorly with the size of the graph. Different from these studies, our framework aims to learn game-theoretical relationships in a supervised manner, while also leveraging the structural symmetries of this problem.

Finally, there has been a recent stream of literature in learning network games from actions of players (Irfan & Ortiz, 2011; Honorio & Ortiz, 2015; Ghoshal & Honorio, 2017a;b; Garg & Jaakkola, 2016; 2017; Barik & Honorio, 2019; 2020; Leng et al., 2020). Most of these methods focus on either

---

[4]For a given player $i$, using the expanded actions $\mathbf{y}_{\cdot k}$ rather than input scalars $x_{\cdot k}$ allows the attention mechanism to produce attention scores which are not necessarily linearly dependent on the value of neighbors action $x_{jk}$, thus producing richer attention scores (see Figure 12 in Appendix).

a binary or a finite discrete action space. For continuous actions, Leng et al. (2020) formulate an optimization problem to learn the structure and marginal benefits of linear quadric games, while Barik & Honorio (2019) are aimed at inferring the network structure from an action-conforming graphical game. Our work differs from existing methods in the literature in that it does not assume a particular game-theoretic structure. Instead, we build a Transformer-like model that learns a mapping from the equilibrium actions to the network structure of the games without explicit knowledge of the utility functions.

## 5 EXPERIMENTS

Our implementation of *NugGeT* uses the sum $\sum_{k=1}^{K}$ as the permutation-invariant functions $\square$ and $\boxed{\cdot}$, and two different 2-layer MLPs for $\phi$ and $\psi$. We use the Adam optimizer (Kingma & Ba, 2015) with a learning rate of $0.001$, a batch size of $100$ and a patience of $50$ epochs. We did not perform any particular hyperparameter tuning for our method, since we found it to be quite robust to the choice of hyperparameters and perform well with standard choices (see Appendix). In all our experiments, we report the mean and the standard error of the mean over the test graphs. Note that we ignore diagonal elements of the adjacency matrix both for training and evaluation. Since they are always zero, the model could easily memorize them, influencing the metrics. We use an AWS p3.16xlarge machine with 8 GPUs. While the training of our model takes between 5 and 10 minutes on a single GPU, the whole set of experiments conducted in the paper necessitate roughly 4 days of GPU time.

### 5.1 SYNTHETIC DATA

**Data Generation**   We follow the setup in Leng et al. (2020) for generating the synthetic graphs using three different random graph models: Erdos–Renyi (ER), Watts-Strogatz (WS), and Barabasi-Albert (BA). More details and exact parameters for the synthetic data are provided in the Appendix. All the graphs have $N = 20$ vertices in our experiments. For each type of graphs above, we simulate equilibrium actions for linear quadratic, linear influence, and BH graphical games using their respective utility function. For linear quadratic games, once the graphs are constructed, we compute $\beta > 0$ such that the spectral radius $\rho(\beta A)$ varies between $0$ and $1$. For linear influence games, when the adjacency matrix is not invertible we take its Moore-Penrose pseudoinverse. For BH games, we use $\epsilon = 0.2$. The generated actions together with the groundtruth network structure are used to train the model. We use 850 graphs in the training set, 50 graphs for validation and 100 graphs for testing and verify that there is no overlap between them.

**Baselines**   We compare with the following general baselines: *Correlation*, *Anticorrelation*, *Graphical Lasso* (Friedman et al., 2008b) and *DeepGraph* (Belilovsky et al., 2017). We also compare to game-specific baselines: *LinQuadOpt (independent)* (Leng et al., 2020) and *LinQuadOpt (homophilous)* (Leng et al., 2020) for linear quadratic games, and *BlockRegression* (Barik & Honorio, 2019) for BH Graphical Games. A more detailed description of the baselines and how they are tuned is provided in the Appendix.

**Results**   Results for Linear Quadratic Games are reported in Figure 3. Columns are different types of graphs (ER, WS, BA), rows are different values of the marginal benefits smoothness $\alpha$ (0, 1, results on $\alpha = 0.5$ are presented in Figure 5 in Appendix), while the x-axis represents the spectral radius $\rho(\beta A)$. *NuGgeT* is on par or superior to other methods in all scenarios, with *DeepGraph* being the runner-up competitor. We observe the largest gap when $\alpha = 1$, i.e. when the marginal benefits are smooth over the graph. Interestingly, it can be observed in the left-most column that both *NuGgeT* and *DeepGraph* performs well both in the case of strategic complements ($\rho(\beta A) > 0$, neighbors take similar actions) and strategic substitutes ($\rho(\beta A) < 0$, neighbors take opposite actions), whereas other baselines only perform well in one of the two cases.

For Linear Influence Games, we report the results in Figure 4. Each plot corresponds to a different type of graph, and the x-axis represents the benefit smoothness $\alpha$. As expected, all methods improve their performance as $\alpha$ grows, since the actions become smoother over the graph (Section 2.3). Again, *NuGgeT* outperforms the baselines in all scenarios. Interestingly, the performance on WS and ER graphs seems to be much lower than for BA graphs. This can be understood empirically by analyzing the eigenvalues of the normalized adjacency matrix $\mathbf{A}$ for different graphs. For WS and ER graphs,

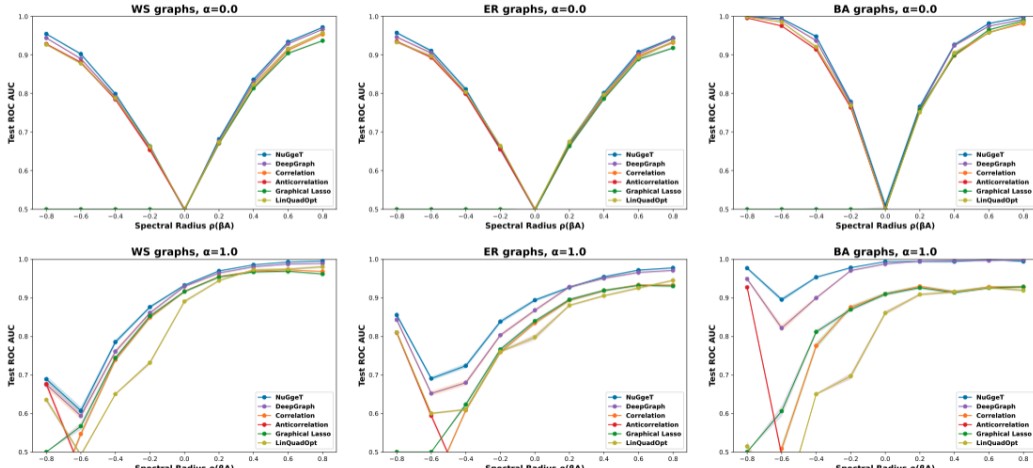

Figure 3: Results for linear quadratic games with varying $\alpha$ and spectral radius $\rho(\beta A)$.

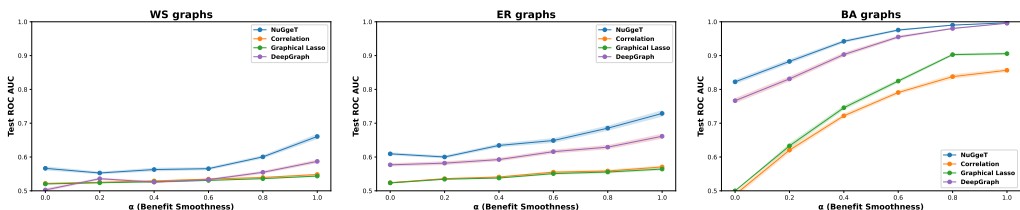

Figure 4: Results on linear influence games when varying the smoothness $\alpha$ of the marginal benefits.

the smallest absolute (non-zero) eigenvalue of $\mathbf{A}$ is on average much smaller than for BA graphs (Figure 7 in Appendix). As explained in section 2.3, eigenvalues with very small absolute values will result in actions behaving like mid-spectrum eigenvectors. We verify this further in Figure 8 in Appendix, which shows the spectral coefficients of the actions for all combinations of graphs and games. Linear Influence games on ER and WS graphs are indeed the only scenario where we empirically observe large coefficients for mid-spectrum eigenvectors. On BA graphs we do not observe this behaviour, and instead mid-spectrum eigenvectors are most often not represented at all by the actions. This is due to BA graphs frequently having exactly zero eigenvalues for mid-spectrum eigenvectors, which are discarded when taking the pseudoinverse.

Results for BH Graphical Games are reported in Figure 6 in Appendix. Since there are no parameters controlling the game, we only have 3 configurations corresponding to different graph types. Again, *NuGgeT* outperforms other methods in all configurations, with the gap being largest on BA graphs. We also perform ablation studies on the number of games, number of training graphs and size of the graphs. The results are presented and discussed in Appendix.

To conclude, the main advantage of our method comes from the ability of learning from data. While learning the mapping from actions to network, the model will learn not only the utility function of the games represented in the data, but also the typical structure of the networks. If all the networks in the training data are Barabasi-Albert graphs, the prediction of the model will likely also be a Barabasi-Albert graph. On the other hand, optimization methods such as LinQuadOpt are not able to automatically bias their prediction to a particular class of graphs.

## 5.2 REAL-WORLD DATA

Players in real-world scenarios often act according to strategic interactions. It has been shown in the sociology and game theory literature that people have an incentive to conform to social norms (Young, 2009; Montanari & Saberi, 2010) or be influenced by their social neighbors (Leng et al., 2020). That is, individual utilities are higher if their behaviors are similar to that of their neighbors in the

social network. Such a mechanism will lead to strategic complement relationships. Following this assumption, we validate our model on two datasets, comparing with the baseline methods introduced in section 5.1.

**Indian Villages**   The *Indian Villages* dataset (Banerjee et al., 2013)[5] contains data from a survey of social networks in 75 villages in rural southern Karnataka, a state in India. Each village constitute a graph, where nodes are households and edges are self-reported friendships. Following the setup in Leng et al. (2020), we consider as actions the number of rooms, number of beds and other decisions families have to make related to their house. We only consider the 48 villages for which we have the ground truth actions and network. 40 are used for training, 3 for validation and 5 for testing. Categorical actions are one-hot encoded, while numerical actions are treated as continuous features. The resulting dataset has graphs with 10 actions and a number of nodes ranging between 77 and 356. It can be seen from Table 2 that *NuGgeT* outperforms all other methods by at least $5.01\%$. *DeepGraph* fails to learn altogether on this dataset.

**Yelp Ratings**   The *Yelp Ratings* dataset[6] consists of rating of users to business, as well as the social connectivity between user. From the raw data we extract 5000 sub-graphs representing communities, where the actions are the average rating of users to 22 categories of businesses. The task is to reconstruct the social connectivity of the users given their actions (ratings). 4250 graphs are used for training, 250 for validation and 500 for testing. More details of the dataset construction are provided in the appendix. On this dataset, NuGgeT outperforms all other baselines by at least $2.79\%$. Overall, the results on real-world data show *NuGgeT* efficacy in cases where the game form is unknown and common assumptions are unrealistic.

| Model | Indian Villages | Yelp Ratings |
|---|---|---|
| *Correlation* | $0.5816\pm0.0135$ | $0.6222\pm0.0043$ |
| *Anticorrelation* | $0.4184\pm0.0135$ | $0.3778\pm0.0043$ |
| *Graphical Lasso* | $0.5823\pm0.0152$ | $0.6523\pm0.0038$ |
| *Baraki and Honorio* | $0.5715\pm0.0164$ | $0.6786\pm0.0032$ |
| *LinQuadOpt (independent)* | $0.5557\pm0.0108$ | $0.6796\pm0.0033$ |
| *LinQuadOpt (homophilous)* | $0.5789\pm0.0174$ | $0.6310\pm0.0036$ |
| *DeepGraph* | $0.4965\pm0.0143$ | $0.6776\pm0.0039$ |
| *NuGgeT* | $\mathbf{0.6324}\pm0.0167$ | $\mathbf{0.7057}\pm0.0035$ |

Table 2: Test ROC AUC for the Indian Villages data and the Yelp Ratings data.

## 6   CONCLUSION AND FUTURE WORK

In this work, we propose a novel framework to infer the network structure behind the games from their equilibrium actions. Unlike existing methods, we achieve so by learning a mapping from the actions to the network structure without knowing the utility function of the game. This is especially beneficial in real-world scenarios where the nature of strategic interactions between players of the game remains hidden or may evolve over time.

**Limitations and Future Work.**   The current work only deals with static games and networks. Moreover, we do not deal with repeated games, i.e. where players have to take multiple actions sequentially. A promising future direction is therefore to extend it to dynamic games and networks, where both the utility function as well as the structure of the network may change over time, as well as dealing with repeated games. In addition, our method cannot guarantee the uniqueness of the learned network; indeed, the main focus of the study is to propose a first and efficient data-driven learning framework without assuming the utility function and prior knowledge about the network structure. We leave the identification of network structure for future work.

---

[5]The Indian Villages dataset can be accessed at https://dataverse.harvard.edu/dataset.xhtml?persistentId=hdl:1902.1/21538
[6]The Yelp dataset can be accessed at https://www.yelp.com/dataset

**Ethics Statement.** Inferring networks of strategic interactions is of key importance to behavior prediction and intervention in network settings. For example, the learned network may be useful for stratification analysis via community detection, or designing efficient targeting strategies in marketing via node centrality analysis. Both would inform decision-making in a wide range of socioeconomic, business, and marketing scenarios.

**Reproducibility Statement.** The code and the data to replicate the experiments are provided as part of the supplementary material. Inside the code folder, the README.md file contains detailed instructions for running the experiments.

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

# A   PROOF OF NUGGET SYMMETRIES

We would like to show that the *NuGgeT* model $g$ is *permutation-invariant* over the set of games and *permutation-equivariant* over the set of nodes.

Starting with the first one, we need to show that $g(\mathbf{X}\mathbf{P}_2) = g(\mathbf{X})$, where $\mathbf{P}_2$ is a $K \times K$ permutation matrix on the games. For what concerns the encoder, we only need to prove that $\alpha_{ij}^{(h)}$ is invariant w.r.t. permutation of the games. Let $p_k$ be the index of the non-zero entry of the $k$-th column of $P_2$, we have:

$$\alpha_{ij}^{(h)} = \boxed{\phantom{x}}_{k=1}^{K} \mathbf{y}_{ik}^{\top} \mathbf{W}_{Q}^{(h)} \mathbf{W}_{K}^{(h)} \mathbf{y}_{jk}$$

$$= \boxed{\phantom{x}}_{k=1}^{K} \mathbf{y}_{ip_k}^{\top} \mathbf{W}_{Q}^{(h)} \mathbf{W}_{K}^{(h)} \mathbf{y}_{jp_k},$$

since $\boxed{\phantom{x}}_{k=1}^{K}$ is chosen to be a permutation invariant operator. This shows that $\alpha_{ij}^{(h)}$ does not depend on the particular permutation $p$ chosen, i.e. it is invariant w.r.t. permutation of the games. Similarly, the same holds for the decoder as $\boxed{\cdot}_{k=1}^{K}$ is a permutation invariant operator and therefore $\boxed{\cdot}_{k=1}^{K} \mathbf{z}_{ik} \odot \mathbf{z}_{jk} = \boxed{\cdot}_{k=1}^{K} \mathbf{z}_{ip_k} \odot \mathbf{z}_{jp_k}$ for any permutation $p$.

Regarding the second part of the proof, we need to show that $g(\mathbf{P}_1\mathbf{X}) = \mathbf{P}_1 g(\mathbf{X})\mathbf{P}_1^{\top}$, where $\mathbf{P}_1$ is an $N \times N$ permutation matrix interpreted as reordering of the nodes of the graph. Letting $p_k$ be the index of the non-zero entry of the $k$-th row of $\mathbf{P}_1$. Note that:

$$\operatorname{enc}(\mathbf{P}_1\mathbf{X})_{ik} = \psi\Big(\mathbf{y}_{p_i k}, \sum_{j=1}^{N} \alpha_{p_i p_j}^{(1)} \mathbf{y}_{p_j k}, \ldots, \sum_{j=1}^{N} \alpha_{p_i p_j}^{(H)} \mathbf{y}_{p_j k}\Big)$$

$$= \psi\Big(\mathbf{y}_{p_i k}, \sum_{j=1}^{N} \alpha_{p_i j}^{(1)} \mathbf{y}_{jk}, \ldots, \sum_{j=1}^{N} \alpha_{p_i j}^{(H)} \mathbf{y}_{jk}\Big)$$

$$= \mathbf{z}_{p_i k}$$

where the second equality stems from the fact that summation is a permutation invariant operator (i.e. $\sum_{j=1}^{N} x_i = \sum_{j=1}^{N} x_{p_i}$ for any permutation $p$). We then have:

$$\Big(\mathbf{P}_1\, g(\mathbf{X})\mathbf{P}_1^{\top}\Big)_{ij} = \hat{a}_{p_i p_j}$$

$$= \psi\Big(\boxed{\cdot}_{k=1}^{K} \mathbf{z}_{p_i k} \odot \mathbf{z}_{p_j k}\Big)$$

$$= \psi\Big(\boxed{\cdot}_{k=1}^{K} \operatorname{enc}(\mathbf{P}_1\mathbf{X})_{ik} \odot \operatorname{enc}(\mathbf{P}_1\mathbf{X})_{jk}\Big)$$

$$= g(\mathbf{P}_1\mathbf{X})_{ij}.$$

Since this holds for all indices $ij$, it follows that $g(\mathbf{P}_1\mathbf{X}) = \mathbf{P}_1 g(\mathbf{X})\mathbf{P}_1^{\top}$.

# B   EXPERIMENTS

## B.1   SYNTHETIC DATA GENERATION

We generate the synthetic data from three different graph models: Erdos-Renyi (ER), Barabasi-Albert (BA) and Watts-Strogatz (WS). In ER graphs, an edge is present with a probability of $p = 0.2$,

independently from all other possible edges. In WS graphs, we set the exact degree of the nodes to be $k = log_2(N)$, with a probability of $p = 0.2$ for the random rewiring process. Finally, in BA graphs, nodes are added one at a time and each new node has $m = 1$ edges which are preferentially attached to existing nodes with already high degree.

## B.2 YELP DATASET GENERATION

Starting from the raw data at https://www.yelp.com/dataset, we create a dataset by performing the following steps:

1. We compute the rating for every user to every businesses category, by averaging the ratings a user has given to all business of each category

2. We weight each edge in the original user-user graph with the fraction of common categories the two users rated at least once

3. We cluster the above weighted graph using Graclus Dhillon et al. (2007) with the objective of minimizing the normalized cut.

4. We discard all the clusters with less than 10 nodes, or with very sparse ratings (less than 25% of the categories with a rating for at least 25% of the users). The result of this step is ∼27k graphs of different users.

5. We rank the clusters extracted above by their density of ratings and keep the 5000 most dense clusters. Each cluster constitutes a graph associated with node attributes, which can be used to train and test our model.

## B.3 BASELINES

In both the synthetic and real-world data experiments we compare with the following baselines:

**Correlation**  The Pearson correlation coefficient between the actions of two nodes. This works particularly well when the actions are homophilous over the graph, i.e. nodes connected in the graph tend to take similar actions. We implement this baseline ourselves.

**Anticorrelation**  The negative of the Pearson correlation coefficient between the actions of two nodes. This works well in case of strategic substitutes, i.e. when nodes connected in the graph tend to take different actions. We implement this baseline ourselves.

**Graphical Lasso Friedman et al. (2008b)**  Computes a sparse penalized estimation of the inverse of the covariance matrix. We use SKGGM[7] QuicGraphicalLasso with empirical covariance initialization for this.

**LinQuadOpt**  Algorithm presented in Leng et al. (2020) which assumes the form of the game to be linear quadratic. It has two version, one where the benefits are assumed to be independent (*LinQuadOpt (Independent)*), and another for homophilous benefits (*LinQuadOpt (Homophilous)*). We implement the algorithm ourselves (no public code was provided).

**BlockRegression**  Algorithm presented in Barik & Honorio (2019), which has been designed specifically for BH graphical games. We re-implement this baseline ourselves following the paper (no public code was provided).

**DeepGraph**  Algorithm presented in Belilovsky et al. (2017) which recovers the graph from the covariance matrix using a series of dilated convolutions. We re-implement *DeepGraph* in PyTorch ourselves taking as inspiration the public TensorFlow implementation of the authors.

All the baselines are tuned on the validation set. For both *Graphical Lasso* and *BlockRegression* we tune the regularization parameter in the range $10^k : k \in [-5, 5]$ with an interval of one, whereas for *LinQuadOpt* both regularization parameters are tuned in $10^k : k \in [-6, 1]$, also with an interval

---

[7]https://github.com/skggm/skggm

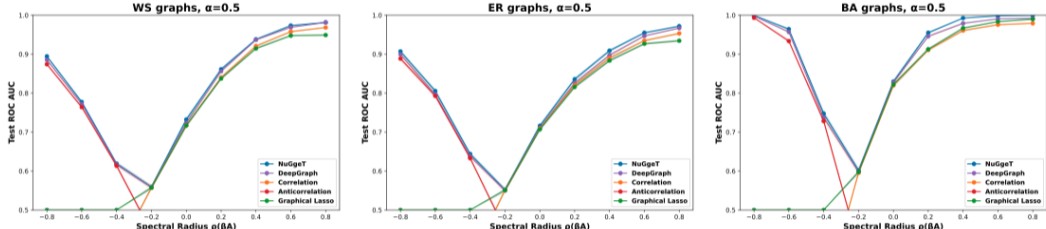

Figure 5: Results for linear quadratic games with $\alpha = 0.5$ and variable spectral radius $\rho(\beta A)$.

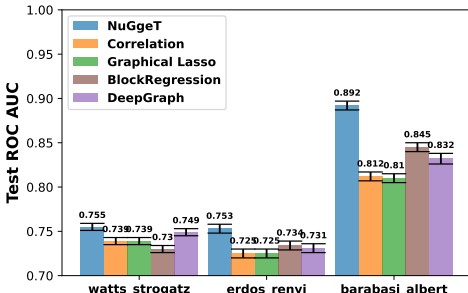

Figure 6: Results for BH graphical games.

of one. We train *DeepGraph* using the same hyperparameters used for NuGget (Adam optimizer, learning rate of 0.001, batch size of 100 and a patience of 50), with $\lceil \log_2(N_{\max}) \rceil$ convolutional layers ($N_{\max}$ corresponds here to the maximum number of nodes of any graph in the dataset) and dilation coefficient equal to $d_k = 2^{k-1}$ for layer $k$ as specified in Belilovsky et al. (2017).

## B.4 NuGgeT Hyperparameters

We do not perform any extensive hyperparameter tuning for the NuGgeT model, but instead use the same standard choice of hyperparameters (reported in Table 3) for all experiments.

| Hyperparameter Name | Value |
|---|---|
| $F$ | 10 |
| $F'$ | 10 |
| $H$ (num of heads) | 10 |
| $\psi$'s num layers | 2 |
| $\psi$'s hidden dim | 100 |

Table 3: Hyperparameters used for *NuGgeT* in all experiments.

## B.5 Additional Results

Additional results on the linear quadratic and BH games are presented in Figure 5 and Figure 6, respectively.

## C Spectral Analysis of the Games

Figure 7 shows that the smallest absolute (non-zero) eigenvalues of the normalized adjacency matrix $\mathbf{A}$ is on average much smaller for ER and WS graphs than for BA graphs. This results in the corresponding eigenvectors having a large influence on the actions for the Linear Influence games, where the Nash equilibrium actions satisfy $\mathbf{x}^* = \mathbf{A}^{-1}\mathbf{b}$. This is confirmed by Figure 8, which shows the graph Fourier coefficients for different eigenvalue indexes: the actions of linear influence games on ER and WS graphs are dominated by mid-range eigenvectors.

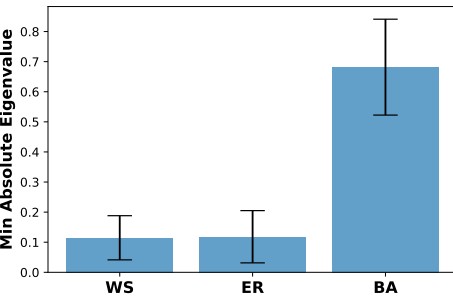

Figure 7: Mean and standard deviation of the minimum absolute non-zero eigenvalue of the adjacency matrix for different types of graph models. Statistics are computed over 1000 graphs with 20 nodes.

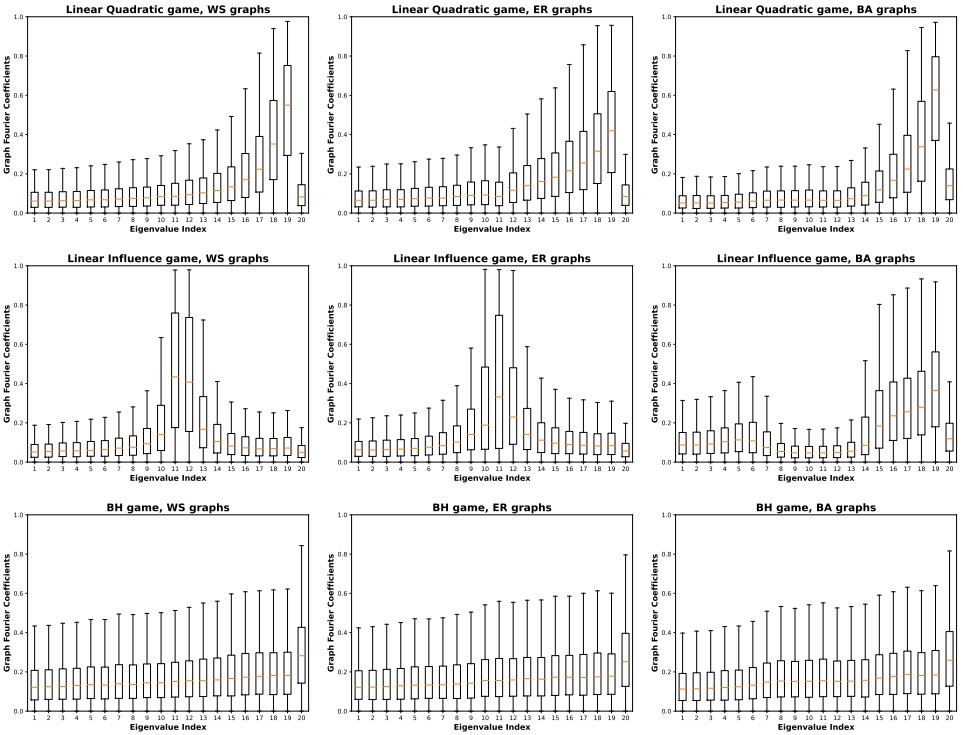

Figure 8: Magnitude of coefficients of normalized action vectors for different games and graphs. These are obtained by taking the Graph Fourier Transform of the actions, i.e. taking the inner product between the action and each of the adjacency matrix eigenvectors.

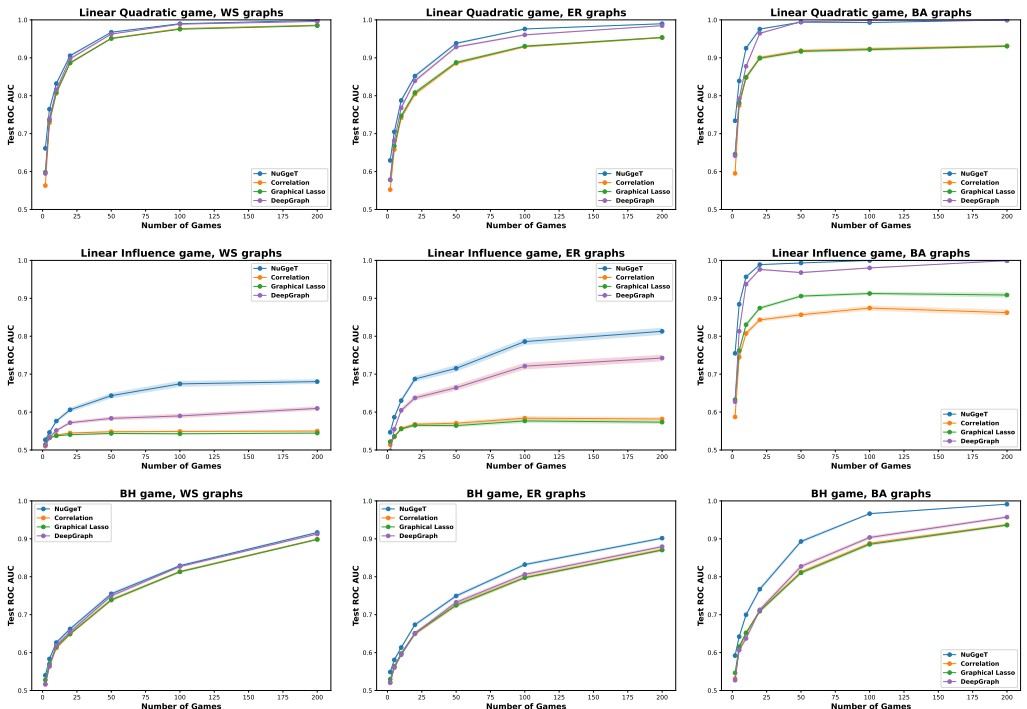

Figure 9: Results with varying number of games.

## D  ABLATION STUDIES

In Figure 9, we investigate how the performance changes with the number of games available. The smaller the number of games, the less information to reconstruct the graph from. We use $\alpha = 1$ for both linear quadratic and linear influence games, and a spectral radius of $0.6$ for linear quadratic games. In line with our expectations, all methods generally improve as more games are available. We analyze the effect of larger graph sizes (Figure 10). The more nodes, the more edge combinations exist and the harder the task becomes, which explains the decrease in performance of both methods as the number of nodes increases. Interestingly, the magnitude of the drop depends heavily on the combination of game and graph types, but *NugGeT* seems to be more robust than *DeepGraph*. We also investigate the effect of the number on training graphs on the model performance (Figure 11). *NugGeT* requires less training graphs compared to *DeepGraph* to obtain a similar performance. Figures 9 and 10 show respectively the performance of various methods when varying the number of games and the number of nodes respectively. We use $\alpha = 1$ for both linear quadratic and linear influence games, and $\beta = 0.6$ (strategic complements) for linear quadratic games. In line with our expectations, all methods generally improve as more games are available. On the other hand, the more nodes, the more edge combinations exist and the harder the task becomes, which explains the decrease in performance of all methods as the number of nodes increases. Interestingly, the magnitude of the drop depends heavily on the combination of game and graph types, but *NugGeT* seems to be more robust than *DeepGraph*.

## E  *NuGgeT*'s ATTENTION MECHANISM

We investigate the difference between a multiplicative attention mechanism operating directly on scalar actions (i.e. with unnormalized attention coefficients of the form: $\tilde{\alpha}_{ij} = x_i x_j a$ with $a$ a learnable parameter), and one operating on transformed actions (i.e. $\tilde{\alpha}_{ij} = \mathbf{y}_i^\top \mathbf{W}_Q \mathbf{W}_K \mathbf{y}_j$, with $\mathbf{y}_i = \text{ReLU}(x_i \boldsymbol{w} + \mathbf{b})$). In Figure 12 we plot the non-normalized attention values of the two different mechanisms for varying values of the input actions $x_i, x_j$. We use a positive and a negative value of $a$ for showing the capability of the simple scalar-based attention mechanism to identify correlation and anti-correlation in the input actions $x_\cdot$ and random coefficients drawn from a normal distribution for $\mathbf{W}_Q, \mathbf{W}_K, \boldsymbol{w}, \mathbf{b}$ for the attention mechanism used in *NuGgeT*. While the scalar attention always

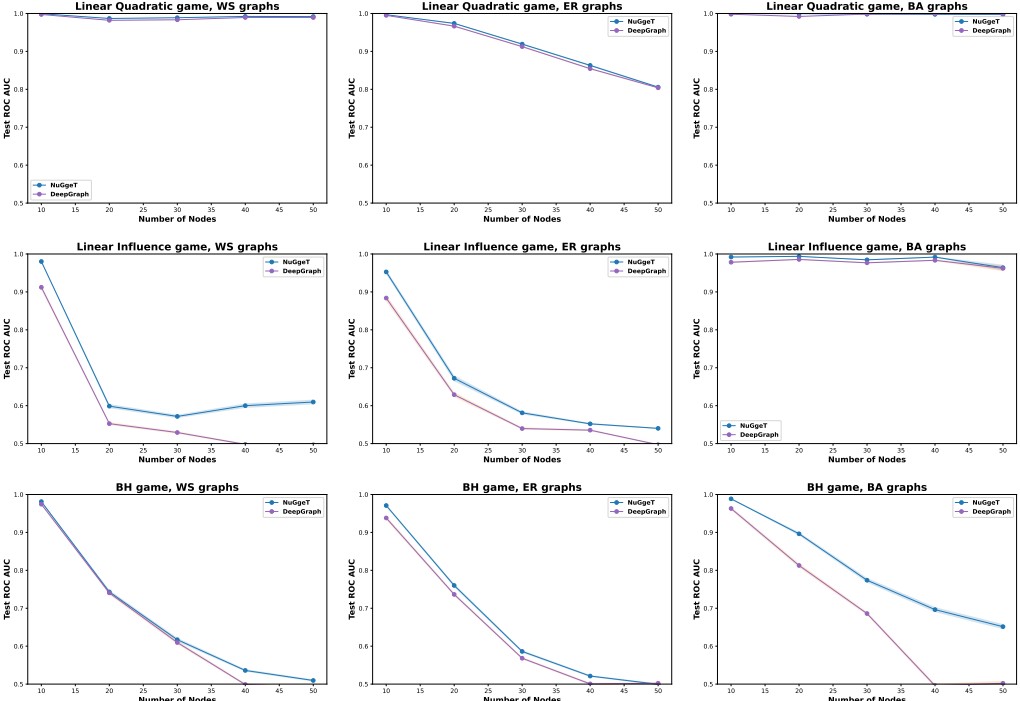

Figure 10: Results with varying number of nodes.

shows a linear trend for a fixed $x_i$ (or $x_j$), the multiplicative layer operating on vectors is able to produce richer non-linear attention patterns with maximum / minimum values for the interior points of $[-1, 1]$.

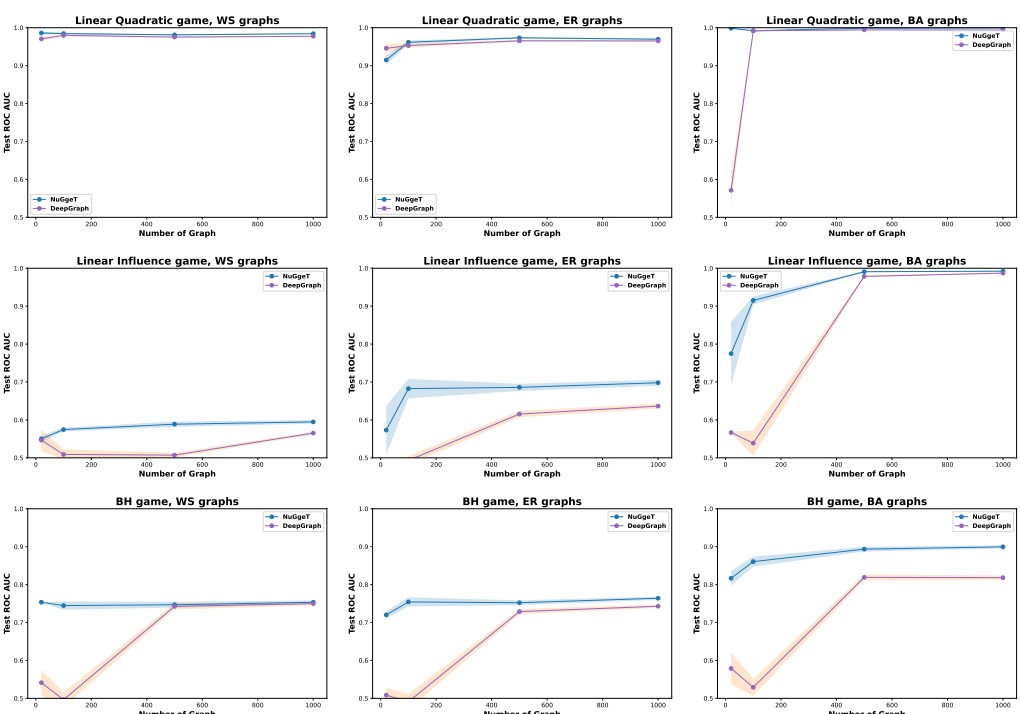

Figure 11: Results with varying number of training graphs.

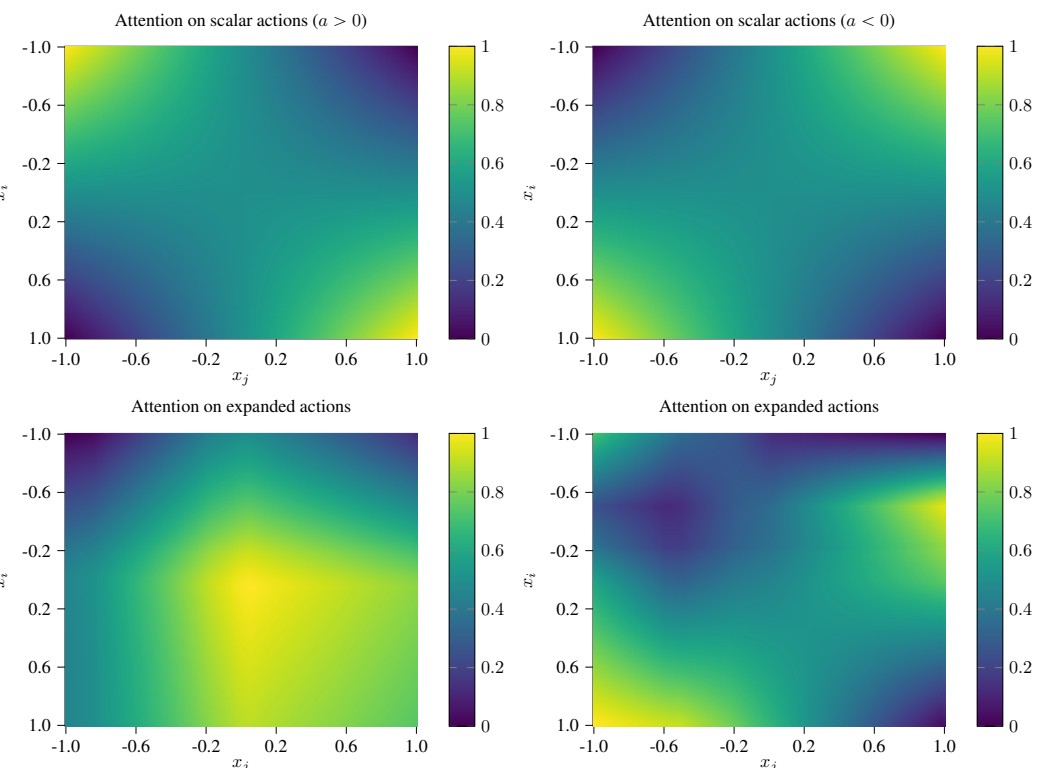

Figure 12: Top, attention values for a multiplicative attention layer directly operating on the input scalars $x_i \in [-1, 1]$, $x_j \in [-1, 1]$ with positive (left) and negative (right) coefficient. Bottom, attention values for a multiplicative attention layer with random coefficients processing the expanded actions $\mathbf{y}_i$, $\mathbf{y}_j$ of $x_i$, $x_j$. The result is scaled to $[0, 1]$ for visualization purposes.

