# OpenReview forum: "Learning to Infer the Structure of Network Games"
_ICLR.cc/2022/Conference — ICLR 2022 Submitted_

### Official Review · Reviewer_awAV · 2021-11-02

**Correctness:** 4
**Technical Novelty And Significance:** 2
**Empirical Novelty And Significance:** 3
**Recommendation:** 3
**Confidence:** 3

**Main Review:**

### Positives

The data-driven approach is novel and some of the architecture design choices made to accommodate invariance properties are well motivated. The problem is well-motivated and the general framework of using just agent actions to infer a network is potentially impactful.

### Weaknesses

However, my two main concerns for the paper are:

### Lack of evidence for good generalization

The authors have shown good results on common synthetic graphs and real-world graphs. However, the train and test datasets are closely related in most cases. For example in the household benchmark, different components are used for train and test. Using subgraphs from the same network or graphs with very uniform properties for ER, WS, and BA graphs for evaluation is not convincing. I would be more convinced if the training and test set has different varieties of graph families so that the model can generalize across multiple graph types.

### The model architecture can be compared with other generative and link prediction models

The proposed model can mostly be replaced by other general network structure inference models including some SOTA generative graph neural models with perhaps some changes in the encoder to preserve the invariance properties. There is nothing particularly special about the problem statement that is leveraged by the model that a slightly modified SOTA generative model can't be replaced with. Therefore, to claim top performance in the proposed task authors may also use at least some of the top Graph generative models [1] and link prediction models [2] as baselines. The current proposed model seems straightforward (transformer encoder + MLP based decoder) and I believe can be improved to leverage ideas from top models.

### Other comments/questions

1. It would be interesting to see what relations the attention heads capture across nodes. Do these model any specific correlation, predict centrality behavior, etc.?
2. In what ways are networks predicted by NuGget different from other baselines? The difference in scores seems small. Therefore, what specific properties are captured by proposed models?
3. Are the scores statistically significant (across multiple runs, train/test splits)?

**Recommendation:** 4 Reject

**Confidence:** 3 Fairly confident.

[1] Guo et al. A Systematic Survey on Deep Generative Models for Graph Generation 2020

[2] Zhang et al Revisiting Graph Neural Networks for Link Prediction 2020

Update: I have read the text of other reviewers. My recommendation doesn't change. I hope the authors improve on the suggestions for submitting their next version.

**Summary Of The Paper:**

The authors introduce a purely data-driven deep learning approach for network structure discovery using just action signals from players of network games with knowing the underlying utility function which can be mostly arbitrary.

The proposed approach uses an encoder that ingests a list of action sets of all users for independent games and outputs a graph (adjacency matrix) over the players. The proposed method outperforms other general network inference baselines as well as baselines that are designed specifically for network games authors evaluate on.

**Summary Of The Review:**

The paper successfully applies Deep graph inference model to the problem of network discovery in network games. The proposed architecture is not very novel and the problem framework, while more challenging, doesn't lend too much to methodological novelty.

With regard to empirical results, I am not convinced of the generalizability and significance of the results. Therefore, I currently recommend rejecting the paper. Providing more extensive evaluation with better baselines and models as well as showing extent of generalizability would strongly help improve the contributions of the paper.

---

### Official Review · Reviewer_BhGi · 2021-11-03

**Correctness:** 4
**Technical Novelty And Significance:** 1
**Empirical Novelty And Significance:** 2
**Recommendation:** 5
**Confidence:** 3

**Main Review:**

Strength of the paper

- As the authors claim, the paper is the first one conducting network inference from equilibrium states of network games without the need of knowing the actual utility functions.

- Converting the network inference problem into a learning framework and applying transformer-like neural architecture contributes to the research community on the network inference task.

Weakness of the paper

- It is unclear that if there is any technical novelty in the paper in constructing the transformer-like architecture for the task. All the components seem to be more or less standard. Permutation invariance may be important for this task, but it is also a technique proposed in some prior work. The full proposed approach of Section 3 is only 1.5 pages, and it is unclear if there is any nontrivial technical novelty, or if any one who is familiar with transformer could come up with a similar architecture and results.

- Related to the above point is that, since the authors claim that their main contribution is by not knowing the utility function in network inference, but would it give better result when knowing the network utility function? It looks like to me that it may be actually hard to incorporate the utility function as the prior knowledge into the transformer-like neural architecture. In other words, by applying the transformer-like architecture, it seems that not knowing the utility function is just a by-product, and not an intentional design point, since anyway it is not easy to incorporate such knowledge into the architecture. If I misunderstood this part, then perhaps the authors should provide an alternative architecture with the utility knowledge and compare the performance difference.

- The experimental evaluation shows that the proposed algorithm performs better, but the margin is not very significant. Even the basic baseline of correlation and anti-correlation performs fairly well, indicating that correlation or anticorrelation explains a large majority of the relationship in the network, at least in the given datasets. Of course, the proposed algorithm performs well on both the positive correlation and negative correlation side. This is not surprising because it learns from the training dataset to detect whether it is positive or negative correlation, while the basic correlation and anticorrelation baseline has no learning phase.

- The experiments are on small network datasets. The synthetic datasets using only 20 nodes are extremely small. Even the ablation study in the appendix (Figure 10) has networks at most size 50. The authors should provide a scalability study to show how large the network size the proposed algorithm can handle, what's the accuracy for large networks, and what's the running time.

Minor comments:

- Page 2, Eq.(1), the notation $a_{ij}$ is not explained. I guess it is the $i$-th row and $j$-column entry of the matrix $A$, but this needs to be clearly stated.

- Page 3, Eq.(8). It is unclear why modeling $b$ as a random vector.

- Page 7. The $y$-axis of the experiment figures are not explained. Even though it may be more or less standard, an explanation is still preferred. A particular thing is that the network is weighted, so it becomes unclear to me what is the ROC AUC in this setting. I would understan ROC AUC when the edges are unweighted, and this it is only a binary classification task on the edges. If the networks used in the experiments are all unweighted, please state this clearly. Moreover, some more tests on weighted graphs are needed.



**Summary Of The Paper:**

The paper studies the problem of inferring the network structure from the equilibrium states of nodes in a network game. The authors put the inference into a learning framework, and it does not need the utility functions of the game. The authors use transformer-like architecture to conduct the learning task. The empirical evaluation on both synthetic networks and real-world networks show that the proposed learning algorithm NuGgeT achieves the best performance comparing to other baselines.


**Summary Of The Review:**

Applying transformer-like architecture to network inference for network games is an interesting direction, but it is unclear how much is the technical contribution of the paper, and the experimental result is not very strong, as discussed above in the weakness part. Therefore, I think the paper is marginally below the acceptance threshold.

---

### Official Review · Reviewer_XNyt · 2021-11-03

**Correctness:** 4
**Technical Novelty And Significance:** 3
**Empirical Novelty And Significance:** Not applicable
**Recommendation:** 6
**Confidence:** 3

**Main Review:**

Strengths:
- Proposing a novel approach to solve the problem of inferring the network structure of games with unknown utility function.
- Justifying/motivating their approach by expressing the equilibrium actions in some network games as a function of network structure.

Comments:
- It seems the assumption of eq (8) for b is central to the proposed method. Because with this assumption the x^* will be probabilistic and it make sense to infer A from a dataset of (X,A). But is it a realistic assumption, i.e., is the real data follows this assumption.
- It would be good if authors discuss about the required no. of data samples to infers the (transformer architecture) model parameters.
- last line of section3: to infer network the structure --> to infer the network structure
- line 9 of section 4: physics --> social or complex networks

**Summary Of The Paper:**

This paper propose a method to infer the network structure behind the games with unknown utility function. Given a dataset of independent games consisting of action and graph (X, G) pairs, and by modeling network structure (adjacency matrix, A) as a (transformer like) function of actions, A = g_{\theta} (X), they inferred the parameters of the model, \theta, by minimize a cross entropy objective function. To justify/motivate their modeling, they show that the PSNE for some well-know (utilities in) networks games, optimal actions, is related to a function of network structure (F(A)) and another function of some additional parameters (b). Then assuming a parametric Gaussian distribution for parameter  b, they derived the multivariate Gaussian distribution of the equilibrium actions x^*.

**Summary Of The Review:**

Proposing a novel approach to solve the problem of inferring the network structure of games with unknown utility function, and in the experiment section authors show the superior performance of their approach compared to other alternative approach.

---

### Official Review · Reviewer_rnrt · 2021-11-04

**Correctness:** 4
**Technical Novelty And Significance:** 2
**Empirical Novelty And Significance:** 3
**Recommendation:** 6
**Confidence:** 2

**Main Review:**

Strengths:
- Clear definitions and paper structure - I'm not very familiar with this line of work but I was still able to follow along.
- Model is reasonably small and takes only minutes to train on a single GPU.

Weaknesses:
- Experiments are conducted on very small graphs (tens of nodes). It is difficult to understand if this would work on much larger networks, and whether the improvements of your model over the baselines would still hold in larger networks.
- Real-world data is tiny or somewhat artificially generated. This might also suggest that the applicability of this method in actuality might not be straightforward. I wish the authors spent more time motivating the problem describing specific examples and cases where it'd be a lot easier to obtain equilibrium action datasets compared to the corresponding network. The scarcity of real-world datasets is making me question the applicability of this problem - if labeled data is hard to get, one alternative would be to provide experiments on unlabeled datasets where meaningful networks are obtained.

**Summary Of The Paper:**

In a network game, players are nodes of a graph and choose their action by maximizing a utility function which depends on their and their neighbors actions, possibly weighted by some given edge weights.

The authors consider the case where each player action is a real number x_i and its payoff is some f(x_i, x_j1, ..x_jm), where j1,..,jm are i's neighbors. The question the authors ask is the following: assuming rationality of the players and observing a noisy version of their actions x_i, can we reconstruct the original graph? In particular, while previous literature has focused on the case where utility function is known, this paper asks whether we can drop this requirement if we observe equilibrium actions for several games (on different networks but with same unknown utility function).

An encoder-decoder model is proposed to learn the mapping from the observed actions to the network structure. A transformer-like architecture is used as encoder to allow for variable-length number of players and permutation-invariance over set of games.

The proposed model performs better than baseline and existing models on synthetic and real-world data.

**Summary Of The Review:**

I'm voting for borderline acceptance for this paper but I'm not very familiar with this area.

---

### Official Review · Reviewer_bS2U · 2021-11-09

**Correctness:** 2
**Technical Novelty And Significance:** 1
**Empirical Novelty And Significance:** 1
**Recommendation:** 3
**Confidence:** 3

**Main Review:**

The paper starts with a significant promise that it can reveal the network structure between agents performing games--- that too without any explicit knowledge of the network. However, later on, I got confused to see that it is a structure learning problem from network data (possibly from steady state of ant networked process). I see this paper has closest connection with information diffusion in networked systems, except that it only estimates parameters from steady state data rather than transient data.
To me, strategic behavior or games is very misleading in the current scenario.


There are other two limitations of the paper:
(1) the paper considers a supervised structure learning setup--- in practice, explicitly networks are rarely observable. I would expect the problem to be practically useful if one tries to find the most likely graph structure which would induce the network data i.e. $X^{\ell}$.
(2) The paper considers same utility function (though somehow I did not find where this utility function is playing a role in the formulation/NN architecture). Such assumption in the context of the current experiments seems a bit impractical. Several subgraphs in Yelp can play different games/strategy.
(3) I did not find the datasets to be very coherent with the main theme of the paper. Is there any strong evidence that people try to strategize in a **collaborative** manner in Yelp? One individual can game the system, but I am not sure there is an underlying https://people.mpi-sws.org/~manuelgr/pubs/netrate-jmlr16.pdf game in the platform.

**Summary Of The Paper:**

The paper aims to infer the structure of the underlying graph where each agent is performing strategically.
Initially it compares three types of games and then it proposes a mechanism that can learn the network without
any functional form of the underlying utility function. Finally, the paper shows the efficacy of the method via experiments.

**Summary Of The Review:**

The paper makes an important promise-- but the formulation and experiments are not coherent with this promise.

---

### Decision · Program_Chairs · 2022-01-20

**Decision:**

Reject

**Comment:**

The paper introduces a transformer-like architecture to perform network inference in network games. While the reviewers acknowledge that the research direction is interesting, they raise concerns regarding the significance of the contribution in terms of methodology, particularly in light of the state of the art, and the experimental evaluation, which in their view did not support the promise of the work. The authors did not reply/follow up on the reviews during the rebuttal period. I would encourage the authors to use the reviewers' comments to revise their paper and resubmit to another conference.